

# Symmetry preference in shapes, faces, flowers and landscapes

Marco Bertamini[1], Giulia Rampone[2], Alexis D.J. Makin[1] and Andrew Jessop[3]

[1] Department of Psychological Science, University of Liverpool, Liverpool, UK
[2] School of Psychology, University of Liverpool, Liverpool, UK
[3] Max Planck Institute for Psycholinguistics, Nijmegen, The Netherlands

## ABSTRACT

Most people like symmetry, and symmetry has been extensively used in visual art and architecture. In this study, we compared preference for images of abstract and familiar objects in the original format or when containing perfect bilateral symmetry. We created pairs of images for different categories: male faces, female faces, polygons, smoothed version of the polygons, flowers, and landscapes. This design allows us to compare symmetry preference in different domains. Each observer saw all categories randomly interleaved but saw only one of the two images in a pair. After recording preference, we recorded a rating of how salient the symmetry was for each image, and measured how quickly observers could decide which of the two images in a pair was symmetrical. Results reveal a general preference for symmetry in the case of shapes and faces. For landscapes, natural (no perfect symmetry) images were preferred. Correlations with judgments of saliency were present but generally low, and for landscapes the salience of symmetry was negatively related to preference. However, even within the category where symmetry was not liked (landscapes), the separate analysis of original and modified stimuli showed an interesting pattern: Salience of symmetry was correlated positively (artificial) or negatively (original) with preference, suggesting different effects of symmetry within the same class of stimuli based on context and categorization.

## INTRODUCTION

Symmetry is often mentioned in connection with aesthetics and preference (*Birkhoff, 1933*; *Osborne, 1986*; *Ramachandran & Hirstein, 1999*). Several studies have demonstrated that observers tend to prefer the more symmetrical version of a given stimulus, using both familiar objects and abstract patterns (*Eisenman, 1967*; *Rhodes et al., 1998*). Some evidence for symmetry as a general aesthetic principle comes from cross cultural studies (for a recent review see *Che et al., 2018*). In addition to explicit measures, implicit measures have confirmed an association between symmetry and positive valence (*Makin, Pecchinenda & Bertamini, 2012b*).

In this study, we examined the preference for symmetry using a deliberately heterogeneous set of images; specifically, faces (males and females), abstract shapes

Corresponding author
Marco Bertamini,
m.bertamini@liverpool.ac.uk

(smooth and angular), flowers, and landscapes. The aim was to compare the role of perfect bilateral symmetry in the context of multiple categories, and to relate preference to rating of the symmetry salience of the items. For one category (landscapes) the symmetry was about positions of objects (composition) and therefore this is a special case of symmetry, that is, not within an object.

## SYMMETRY IN SHAPES, FACES, FLOWERS, AND LANDSCAPES: A BRIEF REVIEW

Chatterjee has observed that there is a link between the existence of brain regions selective for processing faces (fusiform face area), bodies (extrastriate body area) objects (lateral occipital complex, LOC) and places (parahippocampal place area), and the fact that these are common elements in works of art (*Chatterjee, 2014*). This, however, raises the question of whether there are general aesthetic principles, such as preference for symmetry, independent of category, or whether aesthetic principles are specific to each category.

The early work on symmetry and preference used abstract configurations (*Birkhoff, 1933*; *Eisenman, 1967*). It is well established that preference for symmetry correlates with the salience of abstract symmetry. *Jacobsen & Höfel (2003)* and *Höfel & Jacobsen (2007)* used black and white patterns and recorded visual evoked potentials. They found preference for reflectional symmetry and strong activation in visual areas. *Makin, Helmy & Bertamini (2018)* compared preference for abstract patterns where salience had been determined by formal measures of perceptual goodness (*Van Der Helm & Leeuwenberg, 1996*), discrimination speed and amplitude of the visual evoked brain response (*Makin et al., 2016*). Results were straightforward: measures of salience correlated with preference ratings. The more perceptually obvious the symmetry, and the larger the neural response, the more people liked it. Moreover, the results were similar in samples from UK and Egypt.

Other work has shown that implicit preference for symmetry is closely related to how quickly symmetry can be discriminated (*Makin, Pecchinenda & Bertamini, 2012a*, *2012b*). Despite this strong link, it is also known that preference for symmetry varies with several factors, in particular with age and sex (*Humphrey, 1997*) and with training (*Eysenck & Castle, 1970*). Although, importantly, symmetry preference is stable in the context of category learning (*Rentschler et al., 1999*). Moreover, *McManus (2005)* has argued that there is always a tension between symmetry and asymmetry, in the sense that the best balance may not be achieved by perfect symmetry.

The literature on the attractiveness of faces is vast (for a review, see *Rhodes, 2006*). The universality of this preference in humans is supported by cross-cultural and developmental studies (*Perrett, May & Yoshikawa, 1994*). Symmetry contributes to beauty for both males and females faces, with evolutionary psychologists highlighting the role of symmetry in signaling mate quality and health (*Gangestad & Thornhill, 1997*; *Watson & Thornhill, 1994*). Evidence in support of the link between facial symmetry and health is mixed (for a negative finding see *Pound et al., 2014*) and even chicken prefer more symmetrical human faces (*Ghirlanda, Jansson & Enquist, 2002*). There is an alternative position that argues that preference for symmetry is a by-product of how information is processed by neural systems (*Enquist & Arak, 1994*; *Enquist & Johnstone, 1997*).

This alternative view has been highlighted recently by Ryan: "preferences for symmetry in sexual traits may have nothing to do with good genes of the courter but more with how the brains of the choosers work" (*Ryan, 2018*, p.68). However, it should be noted that these two positions are not mutually exclusive.

*Little (2014)* has argued that symmetry preference is domain specific and stronger in the case of faces. This is consistent with an evolutionary view claiming that symmetry is an index of the ability of an organism to cope with developmental stress and thus an index of mate quality. In support of this view it has been reported that preference for symmetry in faces is stronger for upright than for inverted faces (*Little & Jones, 2003*). The study by *Little (2014)* compared responses to human faces, primate faces, and abstract art images. They reported strongest preference for symmetry in human faces. However, the faces were discrete objects while the abstract patterns used were patches of color that did not form a single object. It is known that symmetry is more salient in the case of other grouping factors (i.e., within rather that between objects; e.g., *Bertamini, 2010*). Recently, *Vessel et al. (2018)* found strong inter-individual agreement (i.e., shared taste) for faces and to a lesser extent for landscapes, and low agreement for architecture and in particular for artworks. This suggests the possibility that there are different mechanisms, and different factors affecting symmetry preference for different categories of objects. For faces symmetry may be linked to attractiveness, and possibly mate quality. At the other extreme we have bilateral symmetry between objects, such a physical layout of different objects, it is possible that symmetry is not a predictor of preference.

Flowers are interesting stimuli for different reasons. The growth process may explain some of the symmetry present in flowers, just as in leaves and other biological features. In addition, flowers have evolved to be salient visual stimuli and attractive to insects in particular, as this is their function. Bumblebees, for example, show an innate preference for bilateral symmetry in flowers (*Rodriguez et al., 2004*). In a recent study, *Hůla & Flegr (2016)* asked observers to rate the beauty of 52 common wildflowers. Flower with radial symmetry and low complexity were rated as the most beautiful. The authors note that these are more prototypical as flowers. There was also a preference for more colored flowers (in particular for blue). Although most flowers possess a degree of symmetry, just as for human faces, this symmetry is not perfect. It is possible therefore to modify the images to create flowers with perfect bilateral symmetry.

Landscapes have been studied in terms of preference, although in some cases they have been selected as rich stimuli but without a specific interest in what makes landscapes special. Rapid categorization of scenes is possible, for example, in terms of openness, and is probably based on coarsely localized information (*Oliva & Torralba, 2001*) and color (*Oliva & Schyns, 2000*). With respect to preference, the biophilia hypothesis suggests that preference for open space with sparse vegetation and water may reflect the characteristics of the environment familiar to our common African ancestors (*Wilson, 1984*). In one study, *Falk & Balling (2010)* found evidence for this hypothesis by presenting participants with diverse types of landscapes (see also *Orians & Heerwagen (1992)*). Using functional magnetic resonance imaging, *Yue, Vessel & Biederman (2007)* found that viewing scenes independently rated as preferred, was associated with greater activation in the
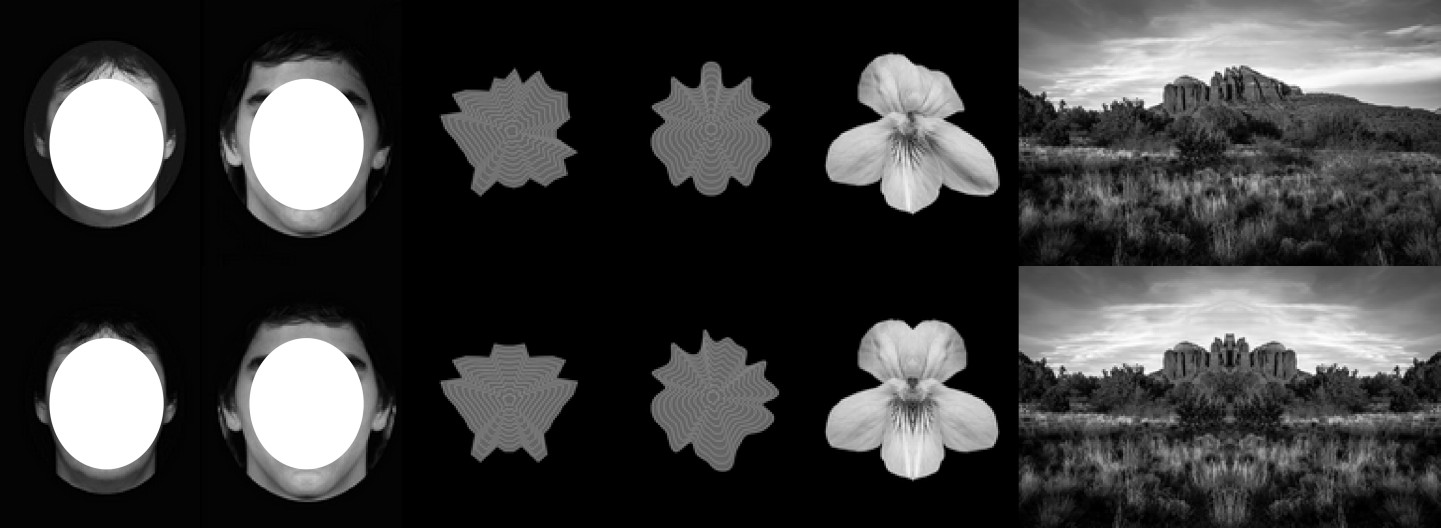

**Figure 1 Examples of images from each of the six categories.** The top row shows the original images that differed in level of symmetry. The second row shows perfect bilateral symmetry. For faces this was a morph of left and right sides and for all other categories was based on the left side only. The unoccluded faces can be seen in the original paper (*Rhodes et al., 1998*).

parahippocampal cortex, but not in the LOC. As in other studies, the category scenes included a variety of settings, some outdoors and some indoors. For the purpose of our study by landscape we mean an outdoor scene with views extending in the distance.

Landscapes are unlike single objects in important ways. A body that is subject to growth will contain some degree of symmetry, but a natural landscape is shaped by multiple physical processes. It is unlikely that hills, boulders, or trees will be located in a symmetrical pattern. This can be contrasted with non-natural, highly constrained scenes of towns or gardens. Our study is therefore the first that allows a direct comparison of the strength of preference for symmetry in landscapes as compared to other categories. As noted above there are good reasons to expect symmetry to be closely associated with objectness, and therefore to have a different function within rather than between objects.

## THE CURRENT STUDY

We selected images of faces (males and females), abstract shapes (smooth and angular), flowers, and landscapes. Observers were asked for ratings of beauty in the context of multiple categories. For each item we had two versions, one had perfect bilateral symmetry and the other had a much lower degree of regularity. We will us the terms original and bilateral symmetry, respectively. Examples of the stimuli are provided in Fig. 1. The images as well as the raw data are available on Open Science Framework (https://osf.io/9qz6p/).

Each observer, however, never saw both versions of a given item. Perfect symmetry is rare in nature, and there is some evidence in the case of faces that when it appears artificial then less symmetrical faces may be preferred in some cases (*Zaidel & Hessamian, 2010*). Therefore, although we predict a preference for symmetry, preference may depend both on its salience (how easy it is to see) and on how artificial it may appear.

In a second phase of the study observers rated how symmetrical each image appeared (rating of symmetry salience) and in a third phase they discriminated which of the pair was the symmetrical version as quickly as possible (response time was recorded). The aim was to test preference for symmetry in a context of multiple categories, without drawing attention to symmetry as the key variable. We will analyze the link between preference and observer ratings of the salience of the symmetry in the pattern. This correspondence could provide support for the view that preference is linked to the tuning of the visual system (*Enquist & Arak, 1994*; *Ramachandran & Hirstein, 1999*; *Redies, 2007*).

People may like what they find easy to perceive, providing a direct link between perception and emotion. *Leder et al. (2004)* formulated a model of visual aesthetic judgment, where symmetry is placed, among other factors, within the early perception analysis stage. Symmetry is an optimal stimulus for the human visual system to process, and observers may like it for this reason. Alternatively, the link between preference and fluency may work in a more indirect way because of the positive associations with fluency of processing. This latter idea is known as fluency hypothesis (*Reber, Schwarz & Winkielman, 2004*). A recent proposal is that fluency as an amplification role, in the sense that it amplify the pre-existing valence of a stimulus (*Albrecht & Carbon, 2014*).

Despite the extensive literature on preference for symmetry, the relationship between salience and preference for symmetry in different domains requires more exploration. There is no guarantee that preference in one domain will generalize to other domains (*Makin, 2017*).

## METHODS

### Participants

A total of 42 individuals (eight male, 34 female) took part in the study and were recruited from the University of Liverpool student community. The age range was 18–24 and six were left handed.

### Stimuli

There were six sets of 10 pairs of images. For each set, 10 images were the original versions (partly symmetrical but without perfect vertical bilateral reflection) and 10 images had perfect vertical symmetry. The sets were: male faces, female faces, angular shapes, smooth shapes, flowers, landscapes. All images were in grayscale. We used a black background and the images appeared within a square region of 500 pixels, except for the landscapes that had a height of 500 pixels and a width of 800 pixels.

Faces: we used, with permission, some of the images from a classic paper on preference for symmetry: *Rhodes et al. (1998)*. There were 10 males and 10 females faces, either as veridical images of human faces or as manipulated images to contain perfect bilateral symmetry, thus generating new images of the 10 individuals.

Abstract shapes: Points were selected from a circle to create a polygon. For more details on the procedure, see *Palumbo, Ruta & Bertamini (2015)* and *Bertamini et al. (2016)*. Around the polygon there were always 20 convex and 16 concave vertices (total 36). The radius of the underlying circle could vary randomly by 54% in length making the

polygon irregular. Symmetrical versions were created by using the left side of the polygon and setting the vertices on the right side by reflection. There were two categories of abstract shapes, one was the original polygons and we refer to these stimuli as Angular (10 pairs). A second set was created by using a cubic spline that made the contour smooth. We refer to this set as Smooth (10 pairs).

Flowers: the flower stimuli were from taken from a study by *Hůla & Flegr (2016)*. In this study, people rated 52 flowers for beauty. Results found a preference for blue color and for radially symmetrical flowers. We selected the following species: Epipactis palustris; Euphrasia rostkoviana; Impatiens noli-tangere; Lathyrus tuberosus; Limodorum abortivum; Melittis melissophyllum; Mimulus moschatus; Ophrys apifera; Pisum sativum; Tropaeolum majus; Veronica beccabunga; Viola biflora; Viola reichenbachiana. Starting from the original images we removed color and manipulated symmetry to obtain 10 pairs of images. One set had the original image of the flower and the other had perfect bilateral symmetry.

Landscapes: Images of outdoor scenes were downloaded from the internet. We used the keyword "landscape" on Google image search with a setting of "free to use share or modify." Starting from the original images we removed color and manipulated vertical symmetry to obtain 10 pairs of images. As for the other categories, one set had the original image of the landscape and the other had perfect bilateral symmetry.

In summary, for all stimuli (with exception for the abstract shapes) symmetry corresponded to an artificial manipulation, as opposed to the original, natural version. This allowed us to test whether the presence of perfect symmetry would automatically predict higher preference, even in objects—and arrangements of objects—that are not perfectly symmetrical by nature.

## Procedure

The experiment had approval from The Health and Life Sciences Research Ethics Committee (Psychology, Health and Society) at the University of Liverpool (Ref: Bertamini: 0540). All participants were given information and signed a consent form before the start of the study.

Each participant was tested individually in a quiet room. They completed three phases always in the same order. First participants rated the beauty of each image using a mouse to control a rating scale (1 = not at all, 10 = very beautiful). Second, they used a similar rating scale to rate how "clear" and "salient" the symmetry was in the images (1 = not at all, 10 = very salient). The question was phrased as follows: "You will be presented with the same images again. You will need to indicate how 'obvious' or 'salient' is the symmetry in each image (from not at all at very salient)."

The last phase had a different format, participants saw a pair of images and selected the more symmetrical of the two. They were asked to respond with one of two keys ("a" and "l") as quickly as possible. The presentation of the stimuli and the recording of the responses was controlled by a program in Python using the PsychoPy software (*Peirce, 2007*). Distance from the screen was not enforced, but at a natural distance of

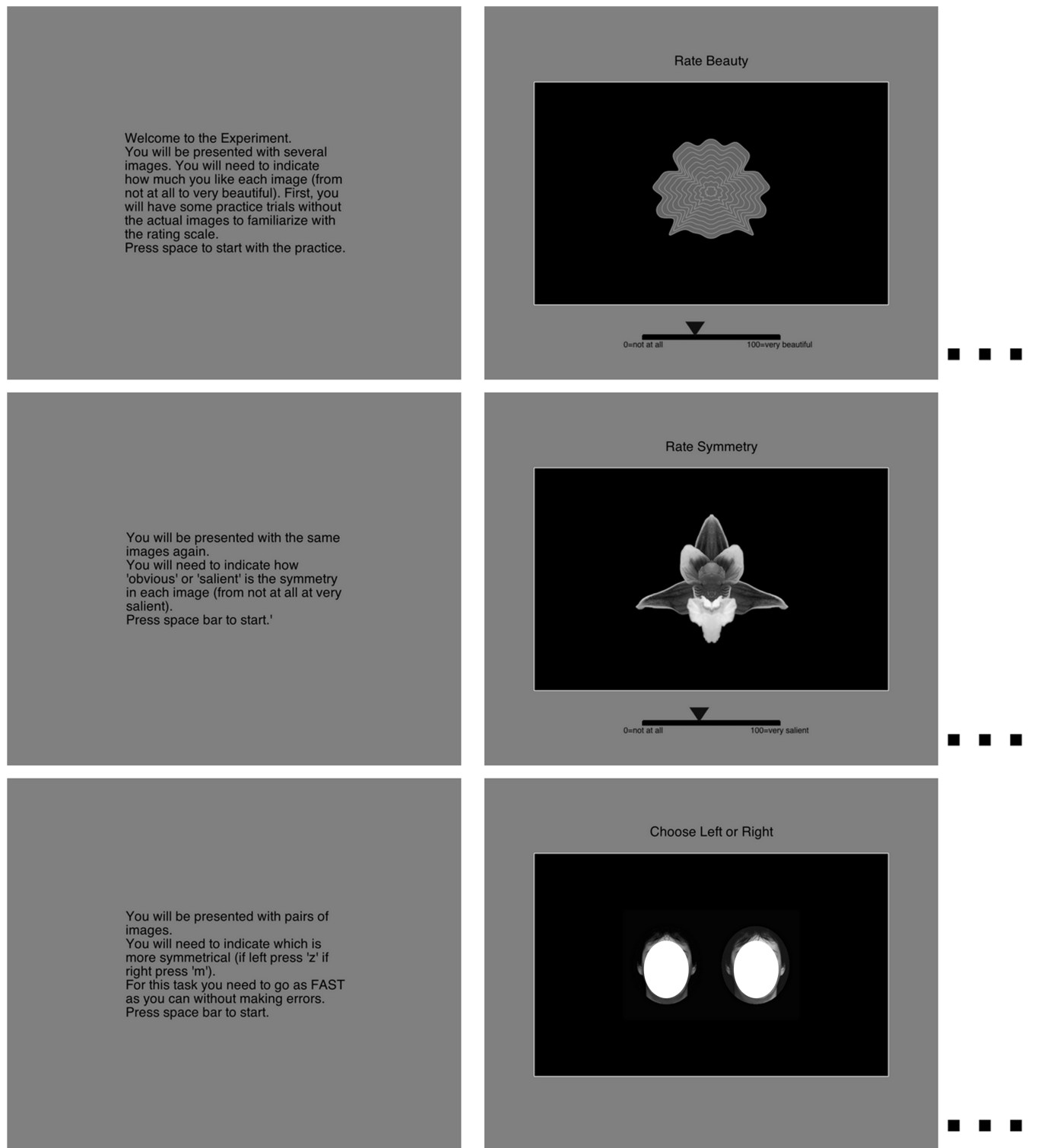

**Figure 2** **An illustration of the three tasks: beauty rating, followed by salience rating, followed by speeded discrimination.** The stimuli were the same for the three tasks but were presented in different random order. The unoccluded faces can be seen in the original paper (*Rhodes et al., 1998*).

approximately 57 cm all stimuli were 10 deg of visual angle in height. The three tasks are shown in Fig. 2.

There were two versions of the procedure. We had 10 pairs of images for each category, one original and one with bilateral symmetry. Therefore, 60 images were selected so that for each category there were five original and five bilaterally symmetrical items. These 60 items were used for one version (A) of the experiment. In another version (B) the total number was the same but for each item we used the symmetrical image (if the original was used in set A) and vice versa. This ensured that for each participant a given face, flower and so on was only presented in one version (either original or symmetrical). This difference between set A and B applies to rating of preference and salience, and not for the discrimination task in which the pair was presented on the screen. Upon completion, participants were asked whether or not they had any strategy throughout the study.

## Analysis

In all the analyses, we split the categories into groups. We tested the results for faces by including male and female as a category. Similarly, we tested the results for abstract shapes by including angular and smooth as a category. This is because we were interested in sex and in angularity as possible factors affecting preference. We did not have similar factors for flowers and for landscapes.

All of the analyses in the present work used generalized linear mixed-effects models implemented in the *lme4* package (*Bates et al., 2015*) in *R* version 3.3.3 (*R Development Core Team, 2018*). The dependent measures of these analyses were subjective ratings of beauty or symmetry, or the response times in the discrimination task. A standardization procedure was applied to these scores, which involved subtracting the mean from the individual ratings, and then dividing each score by the standard deviation. The mean and standard deviations used to standardize the scores were derived from each individual participant rather than the whole sample. All models included symmetry (symmetric/original) as an effect coded fixed factor. Stimulus type was also included in the analyses for faces (male/female) and abstract shapes (angular/smooth), but not for the flower or landscape analysis. The initial random effects structure represented the maximal model (*Barr et al., 2013*). For all analyses, we included participant as a random intercept, together with item (the different images) and version (set A and set B), each with fully crossed random slopes for symmetry and stimulus type. We have no theoretical interest in these variables and we treat them therefore as random factors. The model was simplified until convergence was reached where necessary. Log likelihood-ratio ($\chi^2$) comparisons were obtained through the sequential decomposition of the model (*Bates et al., 2015*), which provided confirmatory tests for the predictors. The marginal and conditional $R^2$ effect sizes are also reported as measures of the variance explained by the model with the random effect structure included (conditional $R^2$) and excluded (marginal $R^2$). from the calculation (*Johnson, 2014*; *Nakagawa & Schielzeth, 2013*; *Nakagawa, Johnson & Schielzeth, 2017*).

## RESULTS

### Preference (beauty ratings)

Before standardization, beauty score was highest for landscapes ($M$ = 74.813; SE = 1.046) and lowest for angular shapes ($M$ = 30.629; SE = 0.884). For smooth shapes it was 33.737 (SE = 0.942). For males and females faces it was, respectively: 32.724, 42.993 (SE = 0.85, 0.971). Finally, for flowers it was 51.578 (SE = 1.1).

Similarly, the standardized beauty score was highest for landscapes ($M$ = 1.343; SE = 0.043) and lowest for angular shapes ($M$ = −0.622; SE = 0.036). For smooth shapes it was −0.476 (SE = 0.038). For males and females faces it was, respectively: −0.498, −0.059 (SE = 0.036, 0.04). Finally, for flowers it was 0.285 (SE = 0.047). Mean values are plotted in Fig. 3.

The maximal model that converged for the face stimuli contained random intercepts for participant, item, and version, but no random slopes. The model revealed that the standardized beauty ratings for faces with perfect bilateral symmetry were significantly higher than for original faces ($\beta$ = 0.0462, SE = 0.0202, $\chi^2$ = 4.26, $p$ = 0.039). Furthermore, there was a significant effect of stimulus type ($\beta$ = −0.2216, SE = 0.0202, $\chi^2$ = 110.7, $p$ < 0.001), as higher ratings of beauty were given to female faces than male faces. There was also a marginal interaction between symmetry and stimulus type ($\beta$ = −0.0354, SE = 0.0202, $\chi^2$ = 3.06, $p$ = 0.080), as a larger decrease in beauty ratings were observed for original female faces than original male faces, compared to their symmetric counterparts. This model accounted for 10.31% of the variance in the data without the random-effects, but 38.93% when they were included ($R_m^2 = 0.1031$; $R_c^2 = 0.3893$).

The maximal model that converged for abstract shapes also only included the random intercepts for participant, item, and version. This revealed that the standardized beauty ratings were significantly higher for symmetrical than original shapes ($\beta$ = 0.1399, SE = 0.0188, $\chi^2$ = 52.09, $p$ < 0.001). Furthermore, there was a significant effect of stimulus type ($\beta$ = −0.0764, SE = 0.0188, $\chi^2$ = 16.34, $p$ < 0.001), as higher ratings of beauty were given to smooth shapes than angular shapes. There was no interaction between symmetry and stimulus type ($\beta$ = −0.0114, SE = 0.0188, $\chi^2$ = 0.37, $p$ = 0.543). This model accounted for 5.37% of the variance in the data without the random-effects, but 40.01% when they were included ($R_m^2 = 0.0537$; $R_c^2 = 0.4001$).

In the case of flowers, the maximal model consisted of the random intercepts for participant, item, and version. This model did not confirm higher beauty ratings for symmetrical flowers ($\beta$ = −0.0155, SE = 0.0357, $\chi^2$ = 0.19, $p$ = 0.664), and accounted for 0.03% of the variance in the data without the random-effects, and 32.51% when they were included ($R_m^2 = 0.0003$; $R_c^2 = 0.3251$). It is worth noting that in the original study by Hůla & Flegr (2016), beauty was mainly linked with radial symmetry of flowers and not bilateral symmetry.

The maximal model for the landscapes data also consisted of random intercepts for participant, item, and version. Conversely, this analysis data showed higher beauty ratings for the original images ($\beta$ = −0.0845, SE = 0.0299, $\chi^2$ = 7.93, $p$ = 0.005) and explained
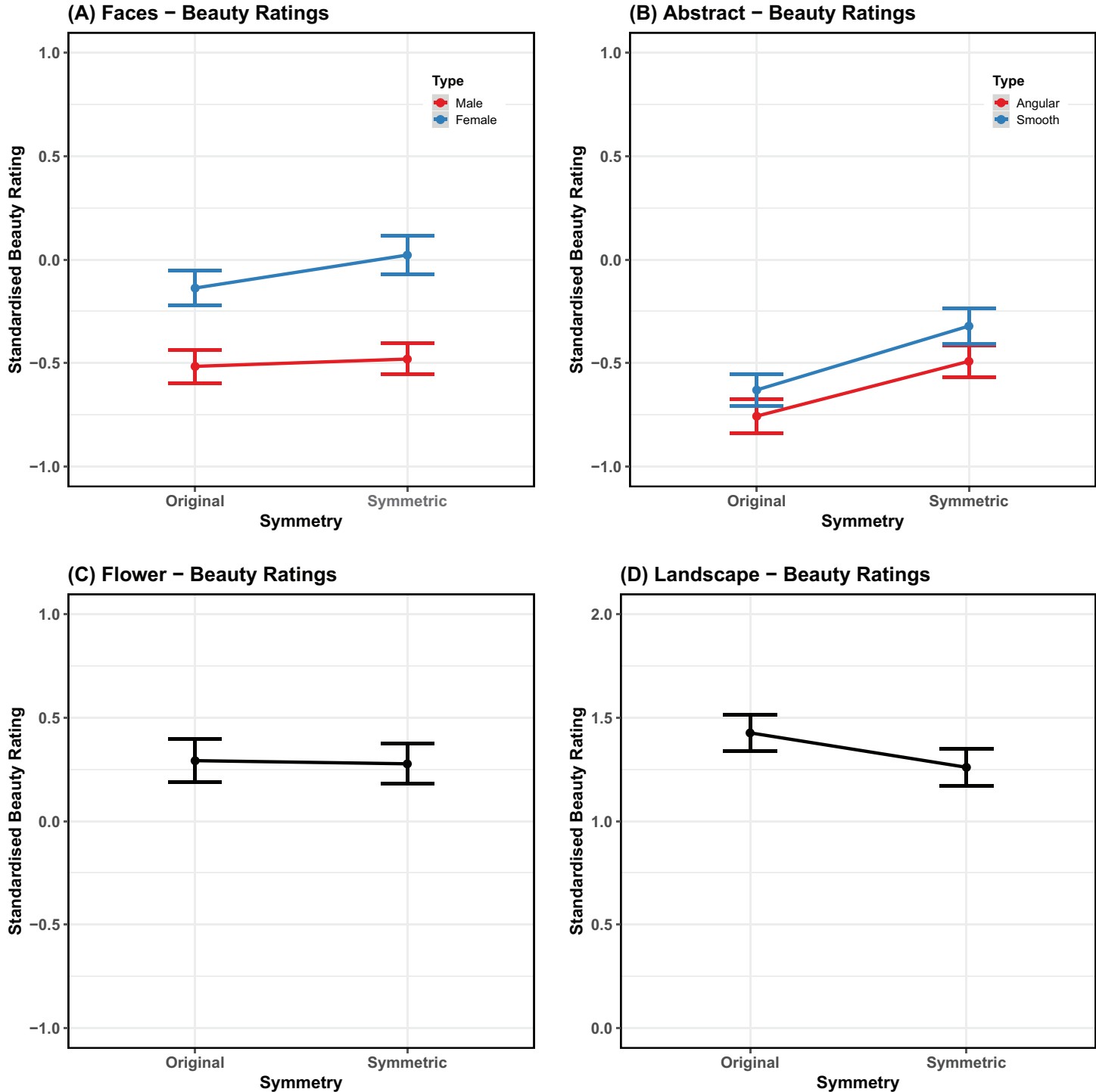

**Figure 3 Beauty ratings.** Mean rating for the beauty task, comparing original stimuli and stimuli with bilateral symmetry (*x* axis). The panels are organized by category and match the way the statistical analyses have been carried out (A) Faces, (B) abstract patterns, (C) flowers, (D) landscapes. Error bars represent the standard error adjusted for the model random effects structure. The scale has the same range in the panels but is shifted upwards in the last one.

1.06% of the variance in the data without the random intercepts and 45.88% when they were included ($R_m^2 = 0.0106$; $R_c^2 = 0.4588$).

## Salience (symmetry ratings)

Ratings of salience of symmetry were analyzed in the same way as ratings for beauty. In all of the models, the maximal random-effects structure that converged included the random intercepts of participant, item, and version, with no random slopes.

The non-standardized salience score was highest for flowers (mean = 66.875; SE = 1.736) and lowest for smooth shapes (mean = 51.496; SE = 2.068). For angular shapes it was 51.924 (SE = 2.023). For males and females faces it was, respectively: 58.247, 61.344 (SE = 1.708, 1.682). Finally, for landscapes it was 58.906 (SE = 2.035). Mean values are plotted in Fig. 4.

The standardized salience score was highest for flowers (mean = 0.267; SE = 0.05) and lowest for smooth shapes (mean = −0.216; SE = 0.06). For angular shapes it was −0.21 (SE = 0.059). For males and females faces it was, respectively: −0.001, 0.089 (SE = 0.049, 0.047). Finally, for landscapes it was 0.036 (SE = 0.06).

The model for faces revealed that the standardized salience ratings were significantly higher for symmetric faces than original faces ($\beta$ = 0.6382, SE = 0.0195, $\chi^2$ = 646.94, $p < 0.001$). Furthermore, there was a significant effect of stimulus type ($\beta$ = −0.0468, SE = 0.0195, $\chi^2$ = 5.9, $p = 0.015$), as higher ratings of symmetry were given to female faces than male faces. There was also an interaction between symmetry and stimulus type ($\beta$ = 0.0645, SE = 0.0195, $\chi^2$ = 10.9, $p < 0.001$), because there was little difference in salience for male and female faces when they were symmetrical, but a larger difference for the original stimuli. This is reasonable given that there was more variability in symmetry in the original faces. This model accounted for 56.73% of the variance in the data without the random-effects, and 60.53% when they were included ($R_m^2 = 0.5673$; $R_c^2 = 0.6053$).

The maximal model for abstract shapes revealed that the standardized beauty ratings were significantly higher for symmetrical than original shapes ($\beta$ = 0.9511, SE = 0.0171, $\chi^2$ = 1,239.09, $p < 0.001$). There was no significant effect of stimulus type ($\beta$ = −6e-04, SE = 0.0171, $\chi^2$ = 1.6e-03, $p = 0.968$) and no interaction between symmetry and stimulus type ($\beta$ = −0.0204, SE = 0.0171, $\chi^2$ = 1.42, $p = 0.233$). This model accounted for 77.21% of the variance in the data without the random-effects, but 79.96% when they were included ($R_m^2 = 0.7721$; $R_c^2 = 0.7996$).

In the case of flowers, the model confirmed higher symmetry salience ratings for symmetrical flowers ($\beta$ = 0.6227, SE = 0.0279, $\chi^2$ = 309.63, $p < 0.001$). This model accounted for 49.74% of the variance in the data without the random-effects, and 63.11% when they were included ($R_m^2 = 0$; $R_c^2 = 0.6311$).

Finally, in the case of landscapes the model revealed higher symmetry salience ratings for symmetrical images ($\beta$ = 0.997, SE = 0.0218, $\chi^2$ = 717.41, $p < 0.001$). This model accounted for 81.92% of the variance in the data without the random-effects, and 83.86% when they were included ($R_m^2 = 0.8192$; $R_c^2 = 0.8386$).

In summary, for all categories more symmetrical images were rated as more symmetrical. People produced ratings of symmetry salience also for images that were

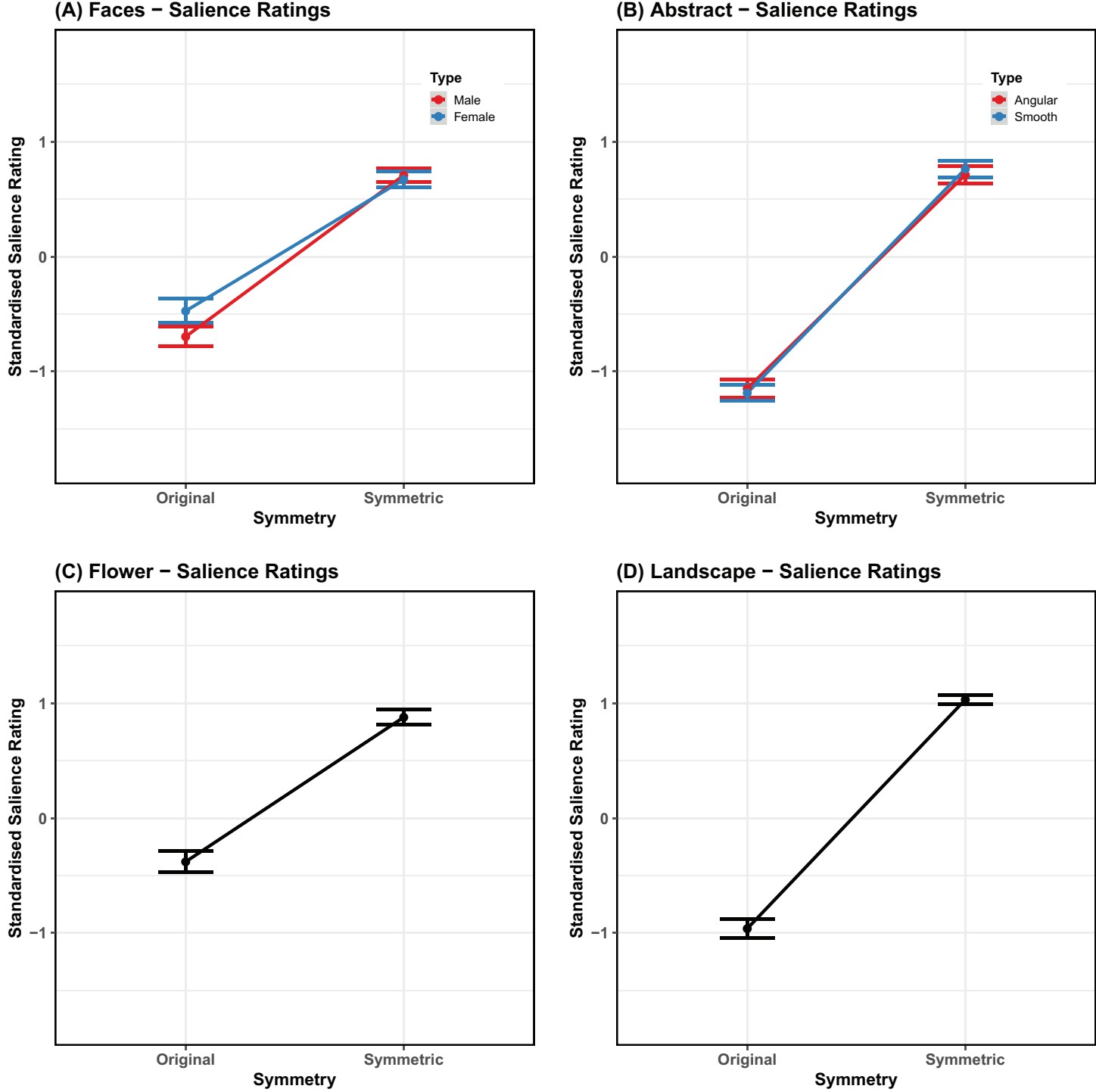

**Figure 4 Salience ratings.** Mean rating for the symmetry salience task, comparing original stimuli, and stimuli with bilateral symmetry (*x* axis). The panels are organized by category and match the way the statistical analyses have been carried out (A) Faces, (B) abstract patterns, (C) flowers, (D) landscapes. Error bars represent the standard error adjusted for the model random effects structure.

perfectly symmetrical. This variability reflects subjective evaluation of symmetry or possible additional symmetry present independently of the bilateral symmetry manipulation.

### Speeded discrimination

The response time necessary to discriminate which of two images was symmetrical was analyzed in the same way as ratings for beauty and salience. The only important difference is that here the factor Position refers to the order of the images in the pair. Position symmetry means that symmetry was on the left (and original on the right) and Position original means that original was on the left (and symmetry on the right). The maximal model that converged contained random intercepts for participant, item, and version, with no random slopes.

Before standardization, response time was fastest for smooth shapes (mean = 0.888; SE = 0.017) and slowest for female faces (mean = 1.294; SE = 0.037). For angular shapes it was 0.893 (SE = 0.019). For males faces it was 1.23 (SE = 0.037). For landscapes it was 1.041 (SE = 0.017) and for flowers it was 1.078 (SE = 0.026). Similarly, the standardized response time was fastest for smooth shapes (mean = −0.231; SE = 0.046) and slowest for female faces (mean = 0.304; SE = 0.067). For angular shapes it was −0.229 (SE = 0.05). For male faces it was 0.066 (SE = 0.054). For landscapes it was 0.068 (SE = 0.049) and for flowers it was 0.046 (SE = 0.058). Mean values are plotted in Fig. 5.

The model revealed that there was no difference in response time when the symmetrical image was presented to the left or to the right ($\beta$ = 0.0263, SE = 0.0388, $\chi^2$ = 0.45, $p$ = 0.503). There was a significant effect of stimulus type ($\beta$ = −0.1187, SE = 0.0388, $\chi^2$ = 9.36, $p$ = 0.002), as faster responses were given to male faces than female faces. There was no interaction between symmetry and stimulus type ($\beta$ = 0.0308, SE = 0.0388, $\chi^2$ = 0.63, $p$ = 0.426). This model accounted for 1.35% of the variance in the data without the random-effects, and 1.73% when they were included ($R_m^2$ = 0.0135; $R_c^2$ = 0.0173).

The model for abstract shapes revealed that there were faster responses when the symmetrical image was presented to the left ($\beta$ = 0.0856, SE = 0.0306, $\chi^2$ = 7.79, $p$ = 0.005). There was no significant effect of stimulus type ($\beta$ = 5.30$e$-05, SE = 0.0306, $\chi^2$ = 1.1$e$-05, $p$ = 0.997) and no interaction ($\beta$ = 0.0471, SE = 0.0306, $\chi^2$ = 2.38, $p$ = 0.123). This model accounted for 1.22% of the variance in the data without the random-effects, but 4.03% when they were included ($R_m^2$ = 0.0122; $R_c^2$ = 0.0403).

In the case of flowers, the model did not confirm any difference whether symmetry was on the left or the right ($\beta$ = 0.0079, SE = 0.0513, $\chi^2$ = 0.02, $p$ = 0.878). This model accounted for <0.01% of the variance in the data without the random-effects, and 5.82% when they were included ($R_m^2$ = 0.00006; $R_c^2$ = 0.0582).

Finally, for landscapes as for flowers there was no difference in response time ($\beta$ = 0.0128, SE = 0.0445, $\chi^2$ = 0.08, $p$ = 0.774). This model accounted for 0.02% of the variance in the data without the random-effects, and 3.62% when they were included ($R_m^2$ = 0.0002; $R_c^2$ = 0.0362).

In summary, for all categories the speeded discrimination was fast and accurate, and differences between categories small. The only additional information was the finding of easier discrimination between symmetrical and original male faces compared to female

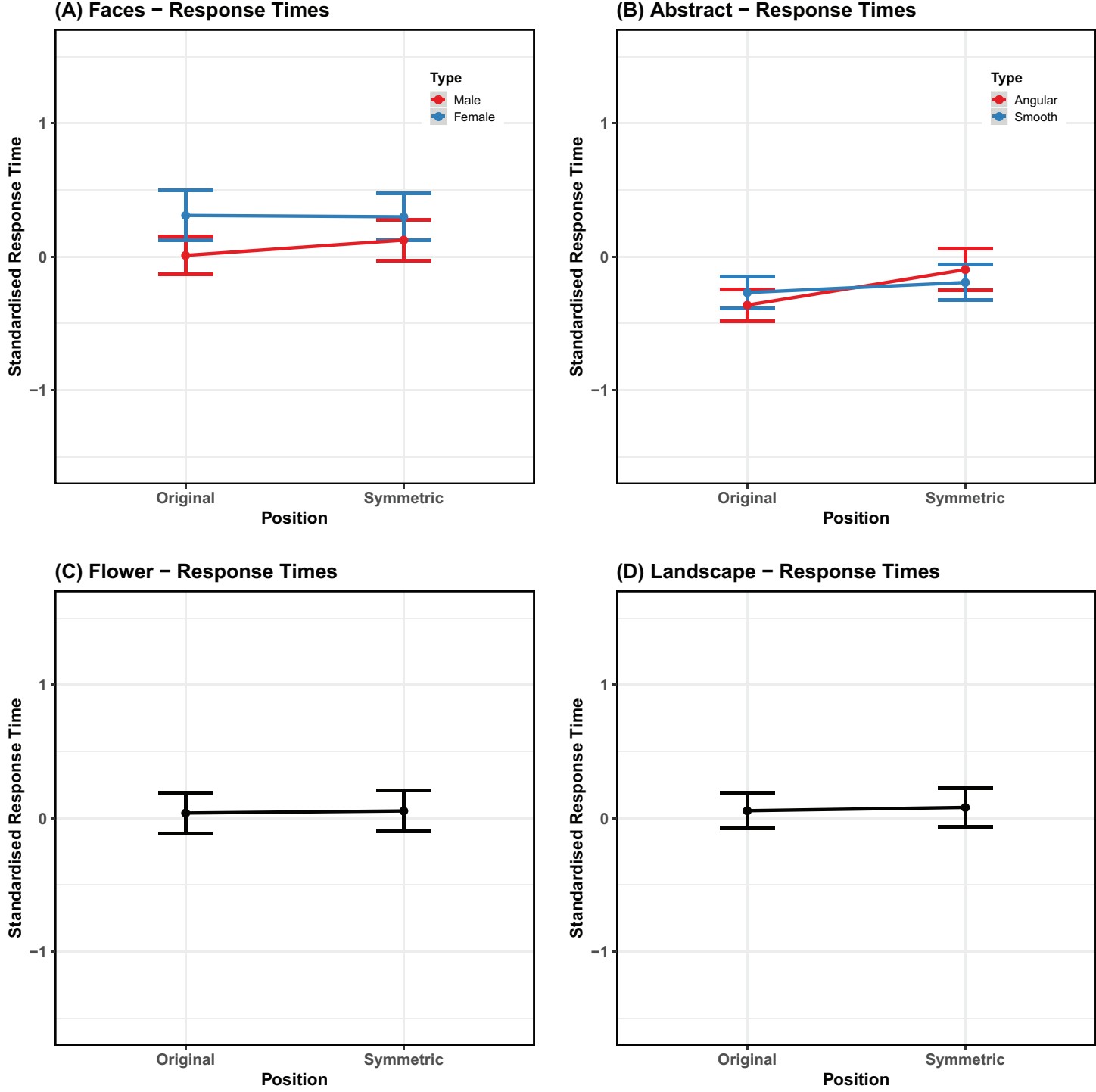

**Figure 5** **Speeded discrimination time.** Mean speeded discrimination task, comparing original stimuli and stimuli with bilateral symmetry (*x* axis). The panels are organized by category and match the way the statistical analyses have been carried out (A) Faces, (B) abstract patterns, (C) flowers, (D) landscapes. Error bars represent the standard error adjusted for the model random effects structure.

faces. This pattern is different from that seen for beauty rating: symmetry affected beauty ratings equally for male and female stimuli, with a trend for a stronger modulation for female rather than male faces.

## Relationship between beauty and salience

We analyzed to what extent the beauty rating reflected the salience of the symmetry in the images. From our first set of analyses we can already see that the category for which symmetry had the strongest effect on beauty rating was that of abstract patterns, and the symmetry for these images was very high. However, symmetry salience was high also for landscapes and here this factor contributed to rating the images as less beautiful. This suggests that symmetry can be used in different ways in different contexts.

Here, we ask a different question. Within each category, were images rated as more symmetrical also the images rated as more beautiful in the same category? Therefore, we used salience rating as a predictor. We analyzed symmetry (original images and images with perfect bilateral symmetry) as a factor. Note that in the first case, symmetry varied among the original objects, such as faces, and in the second case there was perfect bilateral symmetry, but people reported subjective differences in perceived symmetry.

Unlike previous analyses here all the categories are entered in the same analysis. To compare them we used a set of pre-planned contrasts. In particular, we analyzed the difference between landscapes and flowers in one contrast, in another we combined these two to form a new Nature category and compare this to the category of faces. Finally, in another contrast we tested the group of novel stimuli (smooth and angular abstract stimuli) against the familiar objects (faces, flowers, and landscapes). The relationship between beauty and salience ratings is represented graphically in scatterplots in Fig. 6.

For consistency with previous analyses, we entered subject, item, and version as random factors. The model revealed that there was an overall effect of salience in predicting preference ($\beta = 0.1312$, SE $= 0.0383$, $\chi^2 = 5.68$, $p = 0.017$). Other main effects not related to salience were as follows: Beauty ratings were higher for landscapes than flowers ($\beta = -0.448$, SE $= 0.0648$, $\chi^2 = 295.48$, $p < 0.001$), for nature (landscapes/flowers) than faces ($\beta = -0.4584$, SE $= 0.0389$, $\chi^2 = 617.4$, $p < 0.001$), and for real-world stimuli (landscapes/flowers/faces) than abstract shapes ($\beta = -0.2438$, SE $= 0.0221$, $\chi^2 = 549.77$, $p < 0.001$). Many of these main effects also interacted with symmetry and salience. The difference in beauty ratings between abstract stimuli and the other types was smaller for symmetrical than the original stimuli ($\beta = -0.0565$, SE $= 0.0222$, $\chi^2 = 4.3$, $p = 0.038$). Also, increased salience led to a smaller difference in beauty ratings between nature (landscapes/flowers) and faces ($\beta = 0.0364$, SE $= 0.0403$, $\chi^2 = 19.99$, $p < 0.001$), but a larger difference between landscapes and flowers ($\beta = -0.0007$, SE $= 0.0623$, $\chi^2 = 7.62$, $p = 0.006$) and familiar stimuli and abstract shapes ($\beta = -0.0149$, SE $= 0.0211$, $\chi^2 = 9.65$, $p = 0.002$).

Beauty ratings for familiar shapes (faces, flowers, landscapes) were more strongly affected by symmetry than unfamiliar abstract shapes ($\beta = -0.0565$, SE $= 0.0222$, $\chi^2 = 4.3$, $p = 0.038$). Critically, beauty ratings were higher for symmetrical stimuli, but only when this symmetry was salient, as shown by a significant interaction between symmetry and salience ($\beta = -0.1069$, SE $= 0.032$, $\chi^2 = 5.39$, $p = 0.020$). Finally, there was a three-way interaction between salience, symmetry, and the contrast between nature (landscape/flowers) and face stimuli ($\beta = 0.1015$, SE $= 0.0403$, $\chi^2 = 5.78$, $p = 0.016$), which appears to be due to beauty ratings increasing for faces as the salience of their symmetry increases, but
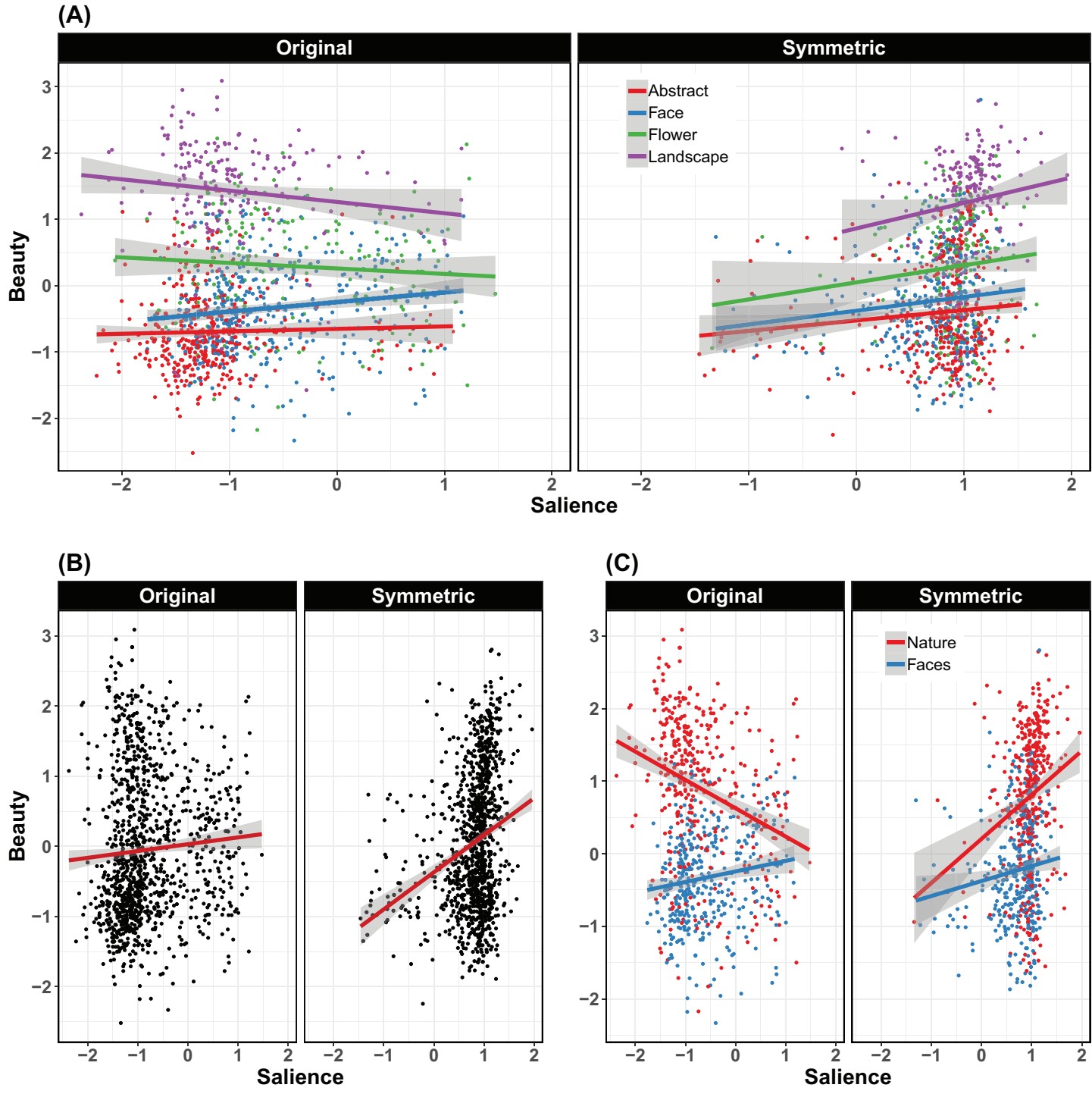

**Figure 6 Relationship between ratings of symmetry and rating of beauty.** Relationship between ratings of symmetry (salience) and rating of beauty. The regression lines match the way the statistical analyses have been carried out. In all graphs, the left panel is for the original stimuli and the right panel for stimuli with bilateral symmetry. The main graph (A) shows all four categories using different colors. To highlight some important relationships the two additional panels combine some of the data. (B) Shows the overall link between salience and beauty. (C) Shows how this relationship is different in different categories and in particular for the nature stimuli (flowers and landscapes).

increasing or decreasing with salience for the nature items depending on symmetry. This model accounted for 47.22% of the variance in the data without the random-effects, and 50.15% when they were included ($R_m^2 = 0.4722$; $R_c^2 = 0.5015$).

These effects are hard to visualize. Therefore, in Fig. 6 we also present two additional plots. These highlight in particular how symmetry salience could be linked in very different ways to beauty rating depending on which set of stimuli we are analyzing: the original or the symmetrical set.

## SYMMETRY PREFERENCE IN INDIVIDUALS

In the previous section, we analyzed whether the individual images that were perceived as more symmetrical were also rated as more beautiful. Here, we turn to the question of whether, across the stimuli, individuals who liked symmetry did so in a consistent manner. Specifically, we examined whether beauty ratings for symmetrical and original stimuli were consistent across the different categories. For each individual, we computed average beauty response for each category, and separately for the symmetrical and original items. These were different items, but within a category we took the difference between the two averages. Therefore, the new value represents how much more the symmetrical stimuli were rated compared to the original stimuli (per person and per category).

Figure 7 shows a scatterplot matrix in which each dot is a person. The two axes are two categories, and thus people in the top right quadrants liked symmetry in both categories (say, they liked both symmetrical faces and symmetrical landscapes). People in the bottom left quadrants disliked symmetry in both categories, and in the other two quadrants we have inconsistent liking responses.

Table 1 shows the correlations that correspond to the patterns of Fig. 7. Both the table and the figure show weak consistency in what individuals liked. In many cases they liked symmetry for one category but not for another. Some of these is not surprising, for example, comparing abstract shapes and landscapes many participants are in the top left quadrant because they disliked symmetry in landscapes and liked it in abstract shapes. Some other cases are more surprising, like the total absence of consistency for symmetry in faces and in flowers. Here liking symmetry in one category had no relationship with liking symmetry in the other.

In the Supplementary Materials we also report a second smaller study ($N = 25$). The procedure and the aims were the same but there were two important changes. One was that in this study observers saw both the original and the bilateral symmetry versions of the items. The other change was that we included color version of the images in addition to the grayscale versions. The results were consistent with the results of our main study, thus supporting the generality and robustness of the findings.

## CONCLUSIONS

Human preference for symmetry and regularity is a well-known phenomenon. In this study, we used a simple rating task with images that were manipulated to introduce perfect bilateral symmetry around the vertical axis. Vertical bilateral symmetry is the most salient type of symmetry (Royer, 1981), and is a non-accidental propriety associated with

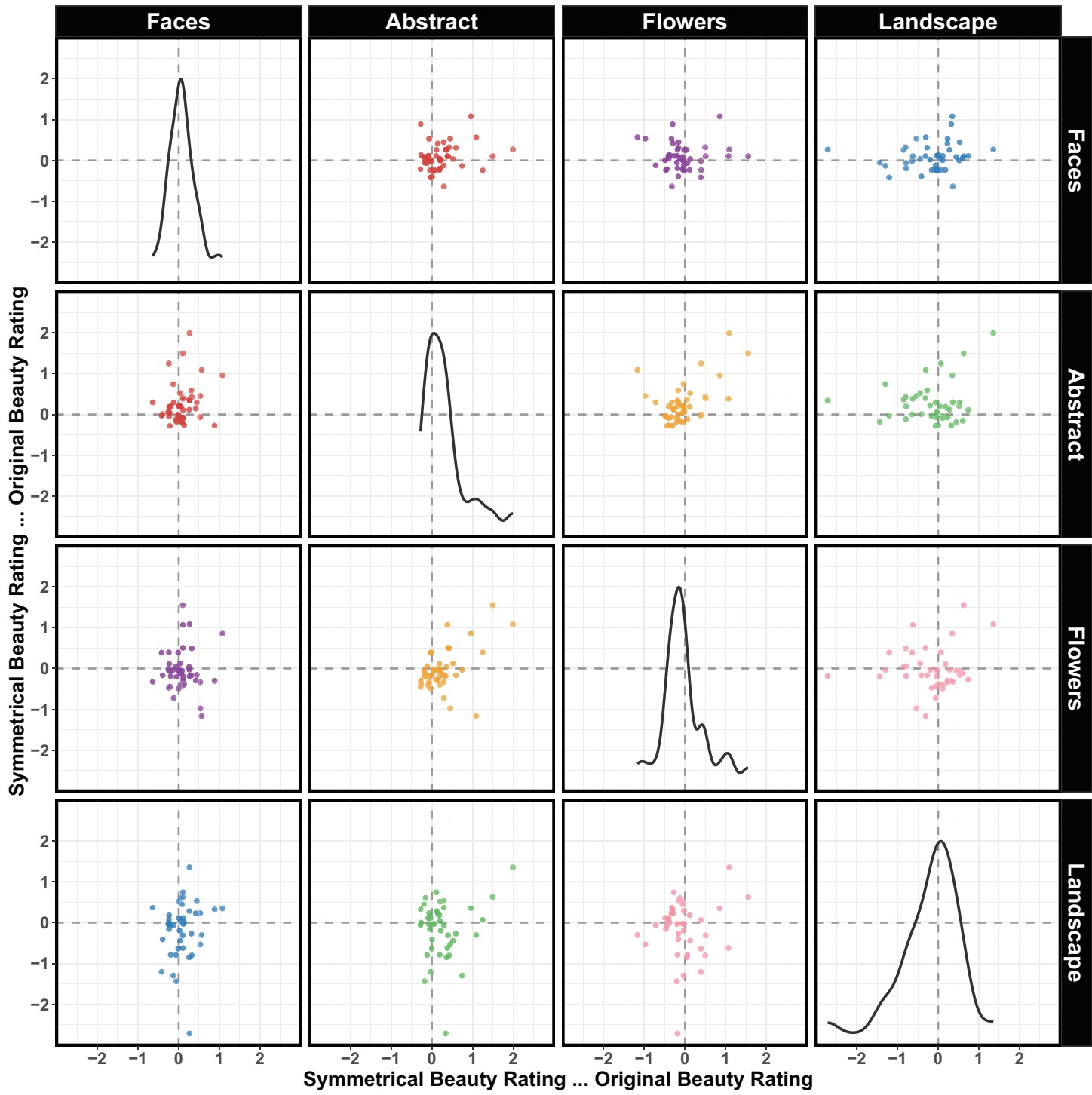

**Figure 7 Relationship of preference in one category and preference for another category.** Scatterplot matrix to show how preference for symmetry for one category relates to preference for another category. Preference for bilateral symmetry is measured as the difference Symmetry—Original. Each point is an individual. People in the top right quadrants liked symmetry in both categories, people in the bottom left quadrants disliked symmetry in both categories, and in the other two quadrants we have inconsistent liking responses.

**Table 1 Correlations for symmetry preference across participants.**

| Stimuli type | Term | Beta | SE | t | p | $R^2$ |
|---|---|---|---|---|---|---|
| Faces | (Intercept) | 0.05 [−0.08–0.18] | 0.07 | 0.69 | = 0.492 | 0.0703 |
| | Abstract | 0.18 [−0.13–0.49] | 0.16 | 1.12 | = 0.264 | |
| | Flowers | −0.08 [−0.37–0.16] | 0.14 | −0.61 | = 0.544 | |
| | Landscape | 0.06 [−0.11–0.19] | 0.08 | 0.72 | = 0.473 | |
| Abstract | (Intercept) | 0.28 [0.15–0.43] | 0.07 | 4.02 | <0.001 | 0.2998 |
| | Faces | 0.28 [−0.21–0.77] | 0.25 | 1.11 | = 0.265 | |
| | Flowers | 0.45 [0.04–0.87] | 0.21 | 2.13 | = 0.033 | |
| | Landscape | 0.07 [−0.14–0.3] | 0.11 | 0.61 | = 0.545 | |
| Flower | (Intercept) | −0.17 [−0.28 to −0.06] | 0.05 | −3.14 | = 0.002 | 0.2581 |
| | Abstract | 0.55 [0.13–1.03] | 0.23 | 2.39 | = 0.017 | |
| | Faces | −0.16 [−0.67–0.36] | 0.26 | −0.61 | = 0.544 | |
| | Landscape | 0.03 [−0.15–0.24] | 0.10 | 0.25 | = 0.801 | |
| Landscapes | (Intercept) | −0.24 [−0.48–0.01] | 0.12 | −1.93 | = 0.053 | 0.0474 |
| | Abstract | 0.19 [−0.4–0.87] | 0.32 | 0.60 | = 0.547 | |
| | Faces | 0.25 [−0.29–0.79] | 0.28 | 0.92 | = 0.359 | |
| | Flowers | 0.06 [−0.34–0.55] | 0.23 | 0.26 | = 0.793 | |

**Note:**
Correlations for symmetry preference across participants.

objectness (*Bertamini, 2010*; *Bertamini, Friedenberg & Kubovy, 1997*). Human faces are a special example of bilateral symmetry and attractiveness ratings increase as symmetry increases (*Rhodes et al., 1998*). In this study, we compared different categories of images, including faces, and also images of landscapes which are not characterized by bilateral symmetry in nature.

For each specific stimulus we created a pair of images, one was the original and the other the symmetrical manipulation (perfect bilateral symmetry). Each observer only saw one item from a given pair. We refer to the two as symmetrical and original, however, original stimuli did possess some degree of symmetry, as is the case for human faces. Therefore, here by symmetrical stimuli we mean stimuli with perfect bilateral symmetry.

We asked what the effect of mixing very different categories of objects would be, and how symmetry preference would relate to symmetry salience in the context of these different categories. We included faces (males and females), abstract shapes (polygons and smoothed polygons), flowers and landscapes. Landscapes are special in that bilateral symmetry here would appear unfamiliar and artificial. The rating for beauty was followed by a second task in which participants rated the salience of symmetry. Finally, they performed a speeded task in which they had to discriminate which of the images in a pair was more symmetrical.

From the ratings of beauty, we can confirm some expected effects. Observers preferred the more symmetrical abstract objects. Independently of symmetry, observers preferred smooth shapes over angular shapes, confirming previous findings for preference for abstract shapes (*Bertamini et al., 2016*). Probably because within abstract shapes the angularity was a salient factor, this effect was strong. Moreover, observers preferred the

more symmetrical faces, in line with previous results (*Little & Jones, 2003*; *Rhodes et al., 1998*). This preference, however, did not apply to flowers. This may appear inconsistent with what reported by *Hůla & Flegr (2016)*. Note, however, that Hůla & Flegr had found a particular preference for flowers with naturally occurring radial symmetry. The symmetry we introduced was an artificial bilateral reflection. Moreover, Hůla and Flegr only used images of actual flowers while we compared original and modified versions of the flowers.

For landscapes the effect of symmetry reversed compared to faces and abstract shapes, and there was a preference for the original landscape over the one with bilateral symmetry. This suggests that context and categorical classification of images are important. There is no doubt that the landscape images had aesthetic value. Most participants rated landscapes as beautiful (compared to the other categories) as shown in Fig. 3. It is likely that introducing perfect symmetry in a natural context was perceived as artificial and unnatural, and consequentially it was disliked. This effect may be similar to that observed for computer generated faces (*Zaidel & Deblieck, 2007*).

The fact that perfect (artificial) symmetry was not preferred in flowers and disliked in landscapes, suggests that symmetry in the image is not a sufficient factor per se to elicit preference, especially in the natural environment. In the case of flowers, the study by *Hůla & Flegr (2016)* found that beauty was related to radial symmetry. They also had ratings of prototypically and therefore they could show that what was judged beautiful was related to what was perceived as the more prototypical image of a flower. For a prototypical wildflower radial rather that bilateral symmetry is the most important factor. In the case of landscapes this was the only category in which symmetry was not associated with an object but rather with a layout of features and objects. In other words, for landscapes symmetry is about composition. However, despite the fact that for landscapes there was a tendency to dislike symmetry, it is remarkable that within the set of symmetrical images of landscapes the higher ratings of beauty were given to those rated as more symmetrical. We consider this surprising result when discussing the correlation between salience and beauty.

Natural landscapes are unlikely to be symmetrical, but often humans manipulate the landscape and introduce regularities. This is clearly the case for landscaped gardens in particular starting with the Italian Renaissance style (Leon Battista Alberti, 1404–1472, wrote a book about architecture that talks extensively about gardens). A recent study, however, has found that the restorative power of gardens was higher for informal rather than formal gardens (*Twedt, Rainey & Proffitt, 2016*).

In a second analysis, we tested ratings of symmetry salience as a predictor of beauty ratings. Overall, we confirmed this link, as it was already observed for faces in the original paper by *Rhodes et al. (1998)*. However, the comparison of the different categories revealed a complex pattern (Fig. 6). While previous work has already established a positive relationship between symmetry salience and symmetry preference for abstract patterns (*Makin et al., 2016*), this did not generalize to landscapes.

The experiment used images in grayscale, and a design that avoided presenting both original and modified versions of an item to a given observer. Color is likely to contribute

to beauty. In a follow-up study we included some of the categories (abstract, flowers, landscapes) in both color and grayscale. We report this experiment in the Supplementary Materials. It confirmed that ratings for beauty were higher in the case of color images, but it also confirmed and supported the other results without interactions with color (see Fig. S1).

In terms of general preference for symmetry in some individuals and not others we did not find much evidence that strong symmetry preference in one category implied strong preference in another category. Individual differences have recently been studied in the context of visual illusions. *Grzeczkowski et al. (2017)* asked the question of common factors in a sample of over 100 participants. If these common factors exist, a person susceptible to one illusion should be susceptible to other illusions as well. They looked at the correlations between the strength of six illusions but could only confirm a correlation between two of them (Ebbinghaus and Ponzo). They came to the conclusion that, in terms of visual perception, individuals are unique.

We highlight two novel aspects of the results. First the role of symmetry varies between categories, as we have seen, and landscape with bilateral symmetry were liked less than the original images of landscapes. We argue that bilateral symmetry is associated with objects and not images with a layout of multiple objects (trees, mountains, etc.). However, this first consideration has to be interpreted keeping in mind a second aspect of the data. Within the original images salience and beauty ratings were negatively correlated. However, within the new symmetrical images of landscapes there was a preference for the more symmetrical images (more precisely those where symmetry was more salient, and ratings of subjective symmetry higher). Although we do not have self-reports on this, it is probable that observers were not aware of this effect of symmetry on their responses, as all images were presented interleaved in a random order. We suggest that symmetry was not related to beauty for landscapes because variation in the layout is a positive quality of a complex landscape. But symmetry was related to beauty for artificial landscapes because these were treated as if they were patterns, or objects, and therefore more similar to the other categories of stimuli.

The context effects that we report are further evidence of how difficult it is to attribute preference to specific factors. Philosophers have expressed skepticism (*Dickie, 1962*), but even within empirical aesthetics problems have been noted (*Holmes & Zanker, 2012*), and one problem is particular is known as the Gestalt nightmare: having a preference for symmetry and for blue does not mean that we can sum these preferences to predict responses to blue symmetrical stimuli (*Makin, 2017*). Although strong claims have been made about the existence of a "single neural currency" for aesthetics (*Skov & Nadal, 2018*), it would be a mistake to conclude from the existence of common reward mechanisms that fixed, objective factors are monotonically contributing to reward, or to preference. Our observed context effects are consistent with other context effects reported. For example, for faces there are both assimilation and contrast effects (*Wedell, Parducci & Geiselman, 1987*). More generally, the nature of human choices is known to be multi-dimensional (*Tversky & Shafir, 1992*). In evaluating visual preference, we found that symmetry can play both a positive and a negative role even within the same study and

for the same category of objects. The critical factor that can reverse the effect of symmetry is its association with an object: bilateral symmetry is expected to be a within-object property and is only judged as beautiful in that context.

### Funding
The authors received no funding for this work.

### Competing Interests
The authors declare that they have no competing interests.

### Author Contributions
- Marco Bertamini conceived and designed the experiments, performed the experiments, analyzed the data, prepared figures and/or tables, authored or reviewed drafts of the paper, approved the final draft.
- Giulia Rampone conceived and designed the experiments, authored or reviewed drafts of the paper, approved the final draft.
- Alexis D.J. Makin conceived and designed the experiments, authored or reviewed drafts of the paper, approved the final draft.
- Andrew Jessop analyzed the data, contributed reagents/materials/analysis tools, prepared figures and/or tables, authored or reviewed drafts of the paper, approved the final draft.

### Human Ethics
The following information was supplied relating to ethical approvals (i.e., approving body and any reference numbers):

The Health and Life Sciences Research Ethics Committee (Psychology, Health, and Society) at the University of Liverpool granted Ethical approval to carry out the study within its facilities (0540).

### Data Availability
Data is available at Open Science Framework: Bertamini, Marco, Giulia Rampone, Alexis D.J. Makin, and Andrew Jessop. 2019. "Symmetry Preference in Shapes, Faces, Flowers, and Landscapes." OSF. January 8. https://osf.io/9qz6p/.

### Supplemental Information
Supplemental information for this article can be found online at http://dx.doi.org/10.7717/peerj.7078#supplemental-information.

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
