# Peer review of "Symmetry preference in shapes, faces, flowers and landscapes"

_PeerJ, doi:10.7717/peerj.7078_

## Round 0.1 · original submission · Major Revisions

Your manuscript has now been seen by 3 reviewers. You will see from their comments below that while they find your work of interest, some major points are raised (in particular by Reviewer 3).

We are interested in the possibility of publishing your study, but would like to consider your response to these concerns in the form of a revised manuscript before we make a final decision on publication. We therefore invite you to revise and resubmit your manuscript, taking into account the points raised. Please highlight all changes in a tracked changes file.

Reviewer 1 ·

Basic reporting

The study's methods and results were clearly articulated. There were a few minor grammatical errors. I would encourage the authors to give the manuscript a careful proof read when resubmitting.

In the paragraph where symmetry as a fitness cue is discussed (starting line 75) it would be worth mentioning conflicting data, from a more recent study (Pound et al. (2014). Proceedings of the Royal Society of London B: Biological Sciences, 281) which calls this account into question.

In the following paragraph (starting line 83) the authors could make clearer the link between Vessel's work on agreement and the current work. Are the authors implying that symmetry is a more salient feature in faces and this accounts for agreement in this context (vs. architecture or artworks)?

The authors may want to consider adding an article in the section on symmetry of flowers, which shows that bumblebees seem to innately prefer bilateral symmetry (Rodríguez et al (2004). Symmetry is in the eye of the ‘beeholder’: innate preference for bilateral symmetry in flower-naïve bumblebees. Naturwissenschaften, 91).

The final paragraph of the results section (starting line 436) would be better placed in the discussion as it is alluding to theoretical considerations of the results. Also, might it not be the case that Grzeckowski et al's (2017) research on illusions suggests that they are different mechanisms underlying these various illusions. This would not necessarily entail that each individual is unique, but more that each illusion is unique. I think Vessel's research on agreement is more pertinent to the question of whether aesthetic taste is individual.

Experimental design

The research question was somewhat clear, however I would have liked to have seen better justification for the categories of stimuli chosen (beyond that they have been the subject of previous research). Why specifically flowers and landscapes? Also, it would be worth making clear from the outset (the first paragraph) why looking across different stimulus categories aids our understanding of the aesthetic value of symmetry.

There were no concrete experimental hypotheses given. While I appreciate this research is somewhat exploratory, I do think on the basis of prior research and with consideration of the conceptual implications of these various stimulus types, the authors could provide predictions of the kind of pattern of results they anticipated.

Validity of the findings

The use of linear mixed effects modelling was effective and patterns in the data are clearly visualised.

·

Basic reporting

In line 46: is stated "positive" response to symmetry on an implicit task. I think would be better to talk about association in that context.

In line 59: when authors talk about "general aesthetic principles" authors must consider to include some more reference to cross cultural research like "Che, J., Sun, X., Gallardo, V., & Nadal, M. (2018). Cross-cultural empirical aesthetics. Progress in Brain Research, 237, 77-103. "

In line 74: Would be appropiate to clarify to which "tensions" is refered.

Figures are correct, but would be better if them showed each dot, maybe changing the transparency of the confidence inteval.

Experimental design

Experimental design is appropiate.

Validity of the findings

Perfectly replicable methods. Conclusions are attached to empirical task.

Additional comments

Would be interesting to take profit of mixed models and instead of comparing group means, use the individua coefficients throught coef(model)$subject. In that way, you are able to compare individual scores for each participant in different tasks.

Reviewer 3 ·

Basic reporting

No comment

Experimental design

(1) The paper requires improvement on conceptual rigour. It uses "symmetry" for "perfect bilateral reflectional symmetry", and "assymmetry" for "non perfect bilateral reflectional symmetry". The authors are well aware that symmetry is a perfectly defined geometrical property, that maybe of many different kinds: rotational, translational, inverse, helical... They also think of this property as "regularity" (in the discussion), without making it clear why. The complementary set, though, can be much diverse, with images classified as "non symmetrical" exhibiting high degrees of, say, rotational symmetry, as it is the case with the flowers set of stimuli. It would be advisable to objectively rate the images for the kinds of symmetry they exhibit in the first place. Images in the "non symmetrical" set may in fact be highly symmetrical. Think of the trisquelion, as an instance.
(2) While the authors take for granted the extensive literature on the influence of symmetry on beauty ratings, what they are interested in is in "symmetry salience" --this is what they ask the participants to rate. Salience in general is a context dependent property of images. Therefore, it is possible for an image to be objectively symmetrical but this symmetry not be salient. As a matter of fact, this is what happens with the flowers images, which are rotational symmetric, but this properties is not as salient as their bilateral symmetry. Some control of factors influencing salience was called for. On the other hand, a look at the stimuli set reveals that bilateral symmetry was present in all of them. Analogously, it can be said that turning coloured images into greyscale ones is not without consequence for salience ratings. Beauty and salience ratings of images in greyscale may not carry over to coloured ones. While for abstract patterns colour may not be particularly important, for flowers colour is probably more salient than any other property. Without control for these effects, the results can be full of confounds.
(3) While the authors acknowledge that perceptual judgments are idiosyncratic, they did not show to participants both images (symmetric and non-symmetric), but just one of them. This opens the possibility that the global results (that in general people prefer symmetric to non-symmetric images, with qualifications), do not hold for each person.
(4) I could not find the instructions given to the participants in the second task: did they mention "symmetry", "bilateral symmetry", "regularity", or something else? What they were asked to tell whether it was salient?
(5) The research question could much more ambitious. It reduces to check whether symmetry (or symmetry salience) influeces in the same way beauty ratings for different kinds of images (landscapes, faces, flowers, abstract patterns. But there is not much theoretical grounding for a hypothesis such as this one, as the authors acknowledge in the discussion.

Validity of the findings

(1) It would be better to report the direct ratings for the first two tasks included, as the sahere the same scale (0-10). Stardardization it is not required for relating both ratings, and it makes the results less intuitive.
(2) The third task -response time- is in a different scale and relating it to the first two dependent variables may require standardization, but it is not carried out in the analysis. It should be better explained what it contributes, given that it does not seem related to the saliency ratings per se.
(3) It would be advisable to proceed backwards: first to validate the set of images to be used as stimuli, then to ask the beauty ratings. It is doubtful that the second and third task provide the validation needed, given that the set of stimuli seem to have been chosed ad hoc.

Additional comments

The authors could be interested in further exploring the notion of regularity -better known in art studies as "rhythm", as the real pull of beauty ratings in naturalist settings, as was already proposed by the Gestalt school. Symmetry (some form of repetion) might just be a proxy for rhythm (repetition with a variation). On the other hand, exploration of non-linear effects of symmetry is also much needed (the influence of symmetry may be sensitive to the presence, or absence, of some other property).

---

## Round 0.2 · accepted · Accept

Thank you for the revised manuscript and response letter. We are delighted to accept your manuscript for publication.

Reviewer 1 ·

Basic reporting

n/a

Experimental design

n/a

Validity of the findings

n/a

·

Basic reporting

All my previous concerns are satisfied

Experimental design

All my previous concerns are satisfied

Validity of the findings

All my previous concerns are satisfied

Additional comments

All my previous concerns are satisfied

Reviewer 3 ·

Basic reporting

Ok

Experimental design

Ok

Validity of the findings

Ok

Additional comments

The paper has been much improved in my view, and it is now fit for publication.